# A permutation-free kernel two-sample test

**Shubhanshu Shekhar**
Department of Statistics and Data Science
Carnegie Mellon University
Pittsburgh, PA 15213
shubhan2@andrew.cmu.edu

**Ilmun Kim**
Department of Statistics and Data Science
Department of Applied Statistics
Yonsei University
ilmun@yonsei.ac.kr

**Aaditya Ramdas**
Department of Statistics and Data Science
Machine Learning Department
Carnegie Mellon University
aramdas@stat.cmu.edu

## Abstract

The kernel Maximum Mean Discrepancy (MMD) is a popular multivariate distance metric between distributions that has found utility in two-sample testing. The usual kernel-MMD test statistic is a degenerate U-statistic under the null, and thus it has an intractable limiting distribution. Hence, to design a level-$\alpha$ test, one usually selects the rejection threshold as the $(1-\alpha)$-quantile of the permutation distribution. The resulting nonparametric test has finite-sample validity but suffers from large computational cost, since every permutation takes quadratic time. We propose the cross-MMD, a new quadratic-time MMD test statistic based on sample-splitting and studentization. We prove that under mild assumptions, the cross-MMD has a limiting standard Gaussian distribution under the null. Importantly, we also show that the resulting test is consistent against any fixed alternative, and when using the Gaussian kernel, it has minimax rate-optimal power against local alternatives. For large sample sizes, our new cross-MMD provides a significant speedup over the MMD, for only a slight loss in power.

## 1 Introduction

We study the two-sample testing problem: given $\mathbb{X} = (X_1, \ldots, X_n) \overset{\text{i.i.d.}}{\sim} P$ and $\mathbb{Y} = (Y_1, \ldots, Y_m) \overset{\text{i.i.d.}}{\sim} Q$, we test the null hypothesis $H_0 : P = Q$ against the alternative $H_1 : P \neq Q$. This is a nonparametric hypothesis problem with a composite null hypothesis and a composite alternative hypothesis. It finds applications in diverse areas such as testing microarray data, clinical diagnosis, and database attribute matching (Gretton et al., 2012a).

A popular approach to solving this problem is based on the *kernel-MMD* distance between the two empirical distributions (Gretton et al., 2006). Given a positive definite kernel $k$, the kernel-MMD distance between two distributions $P$ and $Q$ on $\mathcal{X}$, denoted by $\mathrm{MMD}(P, Q)$, is defined as

$$\mathrm{MMD}(P, Q) = \|\mu - \nu\|_k, \ \text{ where } \ \mu(\cdot) = \int_{\mathcal{X}} k(x, \cdot) dP(x), \text{ and } \nu(\cdot) = \int_{\mathcal{X}} k(x, \cdot) dQ(x). \quad (1)$$

Above, $\mu$ and $\nu$ are commonly called "kernel mean maps", and denote the kernel mean embeddings of the distributions $P$ and $Q$ into the reproducing kernel Hilbert space (RKHS) associated with the positive-definite kernel $k$, and $\|\cdot\|_k$ denotes the corresponding RKHS norm. Under mild conditions on the positive definite kernel $k$ (Sriperumbudur et al., 2011), MMD is a metric on the space of

36th Conference on Neural Information Processing Systems (NeurIPS 2022).

probability distributions. Gretton et al. (2006) suggested using an empirical estimate of the squared distance as the test statistic. In particular, given $\mathbb{X}$ and $\mathbb{Y}$, define the test statistic

$$\widehat{\text{MMD}}^2 := \frac{1}{n(n-1)m(m-1)} \sum_{1 \leq i \neq i' \leq n} \sum_{1 \leq j \neq j' \leq m} h(X_i, X_{i'}, Y_j, Y_{j'}),$$

where $h(x, x', y, y') := k(x, x') - k(x, y') - k(y, x') + k(y, y')$. The above statistic has an alternative form that only takes quadratic time to calculate.

The MMD test rejects the null if $\widehat{\text{MMD}}^2$ exceeds a suitable threshold $\tau \equiv \tau(\alpha)$ that ensures the false positive rate is at most $\alpha$. For "characteristic kernels", this test is *consistent* against fixed alternatives, meaning the power (the probability of rejecting the null when $P \neq Q$) increases to one as $m, n \to \infty$.

The difficulty in practically determining $\tau$ will play a key role in this paper. It is well known that when $P = Q$, $\widehat{\text{MMD}}^2$ is an instance of a "degenerate two-sample U-statistic", meaning that:

$$\text{Under } H_0, \quad \mathbb{E}_P[h(x, X', y, Y')] = \mathbb{E}_Q[h(X, x', Y, y')] = 0.$$

(Above, $x, y, x', y'$ are fixed, and the expectations are over $X, Y, X', Y' \overset{\text{i.i.d.}}{\sim} P$.) As a consequence, its (limiting) null distribution is unwieldy; it is an infinite sum of independent $\chi^2$ random variables weighted by the eigenvalues of an operator that depends on the kernel $k$ and the underlying distribution $P$ (see equation (10) in Appendix A). Since $P$ is unknown, one cannot explicitly calculate $\tau$.

In practice, a permutation-based approach is commonly used, where $\tau$ is set as the $(1 - \alpha)$-quantile of the kernel-MMD statistic computed on $B$ permuted versions of the aggregate data $(\mathbb{X}, \mathbb{Y})$. The resulting test has finite-sample validity, but its practical applicability is reduced due to the high computational complexity; if $B = 200$ permutations are used, the (permuted) test statistic must be recomputed 201 times, rather than once (usually, $B$ is chosen between 100 and 1000).

Due to the high computational complexity of the permutation test, some permutation-free alternatives for selecting $\tau$ have been proposed. However, as we discuss in Section 1.2, these alternatives are either too conservative in practice (using concentration inequalities), or heuristics with no theoretical guarantees (Pearson curves and Gamma approximation) or are only shown to be consistent in the setting where the kernel $k$ does not vary with $n$ (spectral approximation). We later recap some computationally efficient alternatives to $\widehat{\text{MMD}}^2$, but these have significantly lower power.

As far as we are aware, there exists no method in literature based on the kernel-MMD that is (i) permutation-free (does not require permutations), (ii) consistent against any fixed alternative, (iii) achieves minimax rate-optimality against local alternatives, and (iv) is correct for both the fixed kernel setting ($k$ is fixed as $m, n \to \infty$) and the changing kernel setting ($k$ changes as a function of $m, n$, for instance, by selecting the scale parameter of a Gaussian kernel in a data-driven manner).

Our work delivers a novel and simple test satisfying all four desirable properties. We propose a new variant of the kernel-MMD statistic that (after studentization) has a standard Gaussian limiting distribution under the null in both the fixed and changing kernel settings, in low- and high-dimensional settings. There is a computation-statistics tradeoff: our permutation-free test loses about a $\sqrt{2}$ factor in power compared to the standard kernel-MMD test, but it is hundreds of times faster.

**Remark 1.** Let $\mathcal{P}(\mathcal{X})$ denote the set of all probability measures on the observation space $\mathcal{X}$, where we often use $\mathcal{X} = \mathbb{R}^d$ for some $d \geq 1$. For simplicity, in the above presentation, the distributions $P, Q$, kernel $k$ and dimension $d$ did not change with sample size, and this is the setting considered in the majority of the literature. Later, we prove several of our results in a significantly more general setting where $P, Q, d, k$ can vary with $n, m$. Under the null, this provides a much more robust type-I error control in high-dimensional settings, even with data-dependent kernels. Under the alternative, this provides a more fine-grained power result. To elaborate on the latter, we assume that for every $n, m$, the pair $(P, Q) = (P_n, Q_n) \in \mathcal{P}_n^{(1)} \subset \mathcal{P}(\mathcal{X}) \times \mathcal{P}(\mathcal{X})$ for some sequence $\{\mathcal{P}_n^{(1)} : n, m \geq 2\}$. The class $\mathcal{P}_n^{(1)}$ is such that with increasing $n$ and $m$, it contains pairs $(P', Q')$ that are increasingly closer in some distance measure $\varrho$; thus the alternatives can approach the null and be equal in the limit. That is, $\Delta_{n,m} := \inf_{(P', Q') \in \mathcal{P}_n^{(1)}} \varrho(P', Q')$ decreases with $n, m$, and such alternatives are called *local* alternatives (as opposed to *fixed* alternatives). This framework allows us to characterize the *detection boundary* of a test, that is, the smallest perturbation from the null (in terms of $\Delta_{n,m}$) that can be consistently detected by a test.

**Paper outline.** We present an overview of our main results in Section 1.1 and discuss related work in Section 1.2. In Section 2, we present the cross-MMD statistic and obtain its limiting null distribution in Section 2.1. We demonstrate its consistency against fixed alternatives and minimax rate-optimality against smooth local alternatives in Section 2.2. Section 3 contains numerical experiments that demonstrate our theoretical claims. All our proofs are in the supplement.

## 1.1 Overview of our main results

We propose a variant of the quadratic time kernel-MMD statistic of (1) that relies on two key ideas: (i) *sample splitting* and (ii) *studentization*. In particular, we split the sample $\mathbb{X}$ of size $n \geq 2$ into $\mathbb{X}_1$ and $\mathbb{X}_2$ of sizes $n_1 \geq 1$ and $n_2 \geq 1$, respectively (and $\mathbb{Y}$ of size $m \geq 2$ into $\mathbb{Y}_1$ and $\mathbb{Y}_2$ of sizes $m_1 \geq 1$ and $m_2 \geq 1$), and define the *two-sample cross kernel-MMD* statistic $\widehat{\mathrm{xMMD}}^2$ as follows:

$$\widehat{\mathrm{xMMD}}^2 := \frac{1}{n_1 m_1 n_2 m_2} \sum_{i=1}^{n_1} \sum_{i'=1}^{n_2} \sum_{j=1}^{m_1} \sum_{j'=1}^{m_2} h(X_i, X_{i'}, Y_j, Y_{j'}). \tag{2}$$

Our final test statistic is $\widehat{\bar{\mathrm{x}}\mathrm{MMD}}^2 := \widehat{\mathrm{xMMD}}^2 / \widehat{\sigma}$, where $\widehat{\sigma}$ is an empirical variance introduced in (4).

Our first set of results show that quite generally, $\widehat{\bar{\mathrm{x}}\mathrm{MMD}}^2$ has an $N(0,1)$ asymptotic null distribution. Theorem 4 obtains this result in the setting where both the kernel $k$ and null distribution $P$ are fixed. This is then generalized to deal with changing kernels (for instance, Gaussian kernels with data-driven bandwidth choices) in Theorem 5. Finally, in Theorem 15 in Appendix A, we significantly expand the scope of these results by also allowing the null distribution to change with $n$, and also weakening the moment conditions required by Theorem 5.

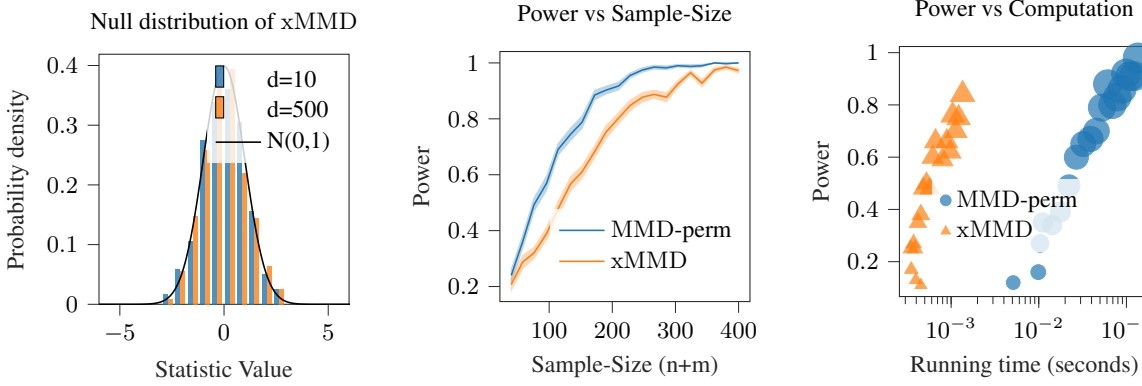

Figure 1: The first figure shows the distribution of our proposed statistic $\widehat{\bar{\mathrm{x}}\mathrm{MMD}}^2$ predicted by Theorem 5 under the null for dimensions $d = 10$ and $d = 100$. The statistic is computed with Gaussian kernel $(k_{s_n}(x, y) = \exp(-s_n\|x - y\|_2^2))$ with scale parameter $s_n$ chosen by the *median* heuristic for different choices of $n$, $m$ and $d$, and with samples $\mathbb{X}$ and $\mathbb{Y}$ drawn from a multivariate Gaussian distribution with identity covariance matrix. The second figure compares the power curves of the two-sample test using the $\widehat{\bar{\mathrm{x}}\mathrm{MMD}}^2$ statistic with the kernel-MMD permutation test (with 200 permutations). The final figure plots the power vs computation time for the two tests. The size of the markers are proportional to the sample-size used in the test.

Our main methodological contribution is the "xMMD test", denoted $\Psi$, which rejects the null if $\widehat{\bar{\mathrm{x}}\mathrm{MMD}}^2$ exceeds $z_{1-\alpha}$, which is the $(1 - \alpha)$-quantile of $N(0,1)$. Formally,

$$\text{xMMD test: } \Psi(\mathbb{X}, \mathbb{Y}) = \mathbb{1}_{\widehat{\bar{\mathrm{x}}\mathrm{MMD}}^2 \geq z_{1-\alpha}}. \tag{3}$$

By the previous results, $\Psi$ has type-I error at most $\alpha$, meaning that $\mathbb{E}[\Psi(\mathbb{X}, \mathbb{Y})] \leq \alpha$ under the null.

We next study the power of the xMMD test $\Psi$ in Section 2.2. First, in the fixed alternative case, i.e., when the distributions $P \neq Q$ do not change with $n$, we show in Theorem 7, that the xMMD test

implemented with any characteristic kernel is consistent under a bounded fourth moment condition. Next, we consider the more challenging case of local alternatives, i.e., when the distributions, $P_n \neq Q_n$, change with $n$. In Theorem 8, we first identify general sufficient conditions for the xMMD test to be uniformly consistent over a class of alternatives. Then, we specialize this to the case when $P_n$ and $Q_n$ admit densities $p_n$ and $q_n$ with $\|p_n - q_n\|_{L^2} \geq \Delta_n$ for some $\Delta_n \to 0$. We show in Theorem 9, that the xMMD test with a Gaussian kernel $k_{s_n}(x, y) = \exp(-s_n \|x - y\|_2^2)$, with scale parameter $s_n$ increasing at an appropriate rate can consistently detect the local alternatives $\{\Delta_n : n \geq 1\}$ decaying at the minimax rate.

Finally, we note that while our primary focus in the paper is on the special case of kernel-MMD statistic, the ideas involved in defining the xMMD statistic can be extended to the case of general two-sample U-statistics. We describe this in Appendix D.1, and obtain sufficient conditions for asymptotic Gaussian limit of the resulting statistic, possibly of independent interest.

## 1.2 Comparisons to related work

**Attempts to avoid permutations.** There have been some prior attempts to avoid permutations, but they are either heuristics (no provable type-I error control) or have poor power (higher type-II error).

The first approach to obtaining a rejection threshold is based on large deviation bounds for the MMD statistic (Gretton et al., 2006, § 3) or the permuted MMD statistic (Kim, 2021, § 5). The resulting tests are distribution-free, but they tend to be too conservative (type-I error much less than $\alpha$, resulting in low power). Another approach involves choosing the threshold based on parametric estimates of the limiting null distribution. For example, Gretton et al. (2006) suggested fitting to the Pearson family of densities based on the first four moments, while Gretton et al. (2009) introduced a more computationally efficient method using a two-parameter Gamma approximation. However, both of these methods are heuristic and do not have any consistency guarantees.

Gretton et al. (2009) introduced a spectral method for approximating the null distribution using the eigendecomposition of the gram matrix. They showed that the resulting distribution converges to the true null distribution as long as the square roots of the eigenvalues associated with the kernel operator are summable. While this method is asymptotically consistent, the conditions imposed on the kernel are more stringent than that used in our work. Furthermore, this method was shown to be consistent only in the fixed kernel (or low-dimensional) setting. Hence, it is unknown whether the results carry over to the case of kernels varying with sample size or high-dimensional settings. This method is also computationally nontrivial due to the need for a full eigendecomposition. Keeping only the top few eigenvectors is another heuristic, but this introduces an extra hyperparameter and loses theoretical guarantees; as a result this method is rarely used in practice. Our methods are simpler (no extra hyperparameter), faster (closed-form threshold), and more robust (type I error guarantees also hold for changing kernels, and in high-dimensional settings).

**Changing the statistic: block-MMD and linear-MMD statistics.** An idea more closely related to ours is that changing the test statistic itself would help make it cheaper to compute and also yield a tractable limiting distribution. One approach splits the observations into disjoint blocks, compute the kernel-MMD statistic on every block, and the final test statistic averages over all the blocks. If the size of each block is fixed, we get a linear-time kernel-MMD (Gretton et al., 2012a,b). The case of block sizes increasing with $n, m$ was studied by Zaremba et al. (2013); Ramdas et al. (2015). Depending on the block size ($b$), the computational complexity of block-MMD statistic varies from linear (constant $b$) to quadratic ($b = \Omega(n)$). Further, if $b = o(n)$, then one gets a Gaussian null distribution as well.

Our proposed statistic is fundamentally different from the block-MMD statistics, despite both being incomplete U-statistics (Lee, 1990). In particular, the block-MMD statistics can be understood as building a block-diagonal approximation of the gram matrix. On the other hand, our proposed cross-MMD statistic uses the *off-diagonal blocks* of the gram matrix, exactly the blocks that the block-MMD with two blocks ($b = n/2$) excludes! The reason that this is a sensible thing to do is nontrivial, and our test is motivated quite differently from the block-MMD. In fact, when $b = n/2$, the block-MMD does not have a Gaussian null, but the cross-MMD does.

For the block-MMD with $b = o(n)$, the Gaussian null distribution is achieved at the cost of suboptimal power, as observed empirically in Zaremba et al. (2013), and proved by Reddi et al. (2015) for the case of linear-MMD and Ramdas et al. (2015) for general block-MMD statistics. In particular, their power is worse by factors scaling with $n$, which means that they are not minimax rate optimal. In

contrast, our test uses exactly half the elements of the gram matrix, and its power is about a $\sqrt{2}$ worse than the MMD test, independent of $n$ and $d$, and we prove explicitly that it achieves the minimax rate.

**Beyond the kernel-MMD.** The literature on two-sample testing is vast, and one can move even further away from the kernel-MMD (than the block-MMD) while retaining some of its intuition. For example, Chwialkowski et al. (2015) proposed a linear-time test statistic by computing the average squared-distance between the empirical kernel embeddings at $J$ randomly drawn points. Jitkrittum et al. (2016) proposed a variant of this statistic in which the $J$ points are selected to maximize a lower bound on the power. In both cases, when the kernel $k$ being used is analytic, in addition to being characteristic and integrable, the authors showed that the limiting distribution under null for this statistic is a combination of $J$ independent $\chi^2$ random variables. However, similar to the spectral method of Gretton et al. (2009), the high-dimensional behavior of these statistics are unknown. In fact, some preliminary experiments with $d \approx n$ in Appendix E.3 suggest that these linear-time statistics have a different null distribution in this regime. Further, the authors only proved consistency of these tests against fixed alternatives, but their power is not known to be minimax rate optimal. In contrast, our statistic has the same limiting distribution in low- and high-dimensional settings even with changing kernels, and is provably minimax rate optimal for smooth alternatives, and it is a much more direct tweak of the usual MMD test.

**One-sample (goodness-of-fit) testing.** Kim and Ramdas (2020) proposed and analyzed a similar studentized cross U-statistic in the simpler one-sample setting. Our work has different motivations: our primary goal in this paper is to design a permutation-free kernel-MMD test that does not significantly sacrifice the power, while Kim and Ramdas (2020) pursued the related but different goal of *dimension-agnostic* inference, which means having the same limiting distribution in low-dimensional and high-dimensional settings. Nevertheless, our results can be seen as an extension of their methods to two-sample testing. Our proofs also build on their advances, but we require a more involved analysis since in their case the second distribution is known (making it a point null).

## 2 Deriving the cross-MMD test

In this section, we present our test statistic and investigate its limiting distribution. First note that the squared kernel-MMD distance between two probability measures $P$ and $Q$ can be expressed as an inner product, namely $\langle \mu - \nu, \mu - \nu \rangle_k$. The usual kernel-MMD statistic is obtained by plugging the empirical kernel embeddings into this inner product expression and removing the diagonal terms to make it unbiased. Our proposal instead considers pairs of empirical estimates $(\widehat{\mu}_1, \widehat{\mu}_2)$ and $(\widehat{\nu}_1, \widehat{\nu}_2)$ constructed via sample splitting, and use the inner product between $\widehat{\mu}_1 - \widehat{\nu}_1$ and $\widehat{\mu}_2 - \widehat{\nu}_2$ instead. This careful construction allows us to obtain a Gaussian limiting distribution after studentization. To elaborate, recall from Section 1.1 that we partition $\mathbb{X}$ into $\mathbb{X}_1$ and $\mathbb{X}_2$, and similarly $\mathbb{Y}$ into $\mathbb{Y}_1$ and $\mathbb{Y}_2$. We then compute empirical kernel embeddings based on each partition, yielding $\widehat{\mu}_1 := n_1^{-1} \sum_{i=1}^{n_1} k(X_i, \cdot)$, $\widehat{\mu}_2 := n_2^{-1} \sum_{i'=1}^{n_2} k(X_{i'}, \cdot)$, $\widehat{\nu}_1 := m_1^{-1} \sum_{j=1}^{m_1} k(Y_j, \cdot)$ and $\widehat{\nu}_2 := m_2^{-1} \sum_{j'=1}^{m_2} k(Y_{j'}, \cdot)$. Using these embeddings coupled with the kernel trick, the cross U-statistic (2) can be written as $\widehat{\text{xMMD}}^2 = \langle \widehat{\mu}_1 - \widehat{\nu}_1, \ \widehat{\mu}_2 - \widehat{\nu}_2 \rangle_k$. To further motivate our test statistic, denote $U_{X,i} := \langle k(X_i, \cdot), \ \widehat{\mu}_2 - \widehat{\nu}_2 \rangle_k$ for $i = 1, \ldots, n_1$ and $U_{Y,j} := \langle k(Y_j, \cdot), \ \widehat{\mu}_2 - \widehat{\nu}_2 \rangle_k$ for $j = 1, \ldots, m_1$. Then the cross U-statistic can be viewed as the difference between two sample means: $\widehat{\text{xMMD}}^2 = \frac{1}{n_1} \sum_{i=1}^{n_1} U_{X,i} - \frac{1}{m_1} \sum_{j=1}^{m_1} U_{Y,j}$. Since the summands are independent *conditional on* $\mathbb{X}_2$ *and* $\mathbb{Y}_2$, one may expect that $\widehat{\text{xMMD}}^2$ is approximately Gaussian after studentization. Our results in Section 2.1 formalize this intuition under standard moment conditions, where it takes some care to remove the above conditioning, since we care about the unconditional distribution.

Let us further denote the sample means of $U_{X,i}$'s and $U_{Y,j}$'s by $\bar{U}_X$ and $\bar{U}_Y$, respectively, and define

$$\widehat{\sigma}_X^2 := \frac{1}{n_1} \sum_{i=1}^{n_1} \left( U_{X,i} - \bar{U}_X \right)^2, \ \widehat{\sigma}_Y^2 := \frac{1}{m_1} \sum_{j=1}^{m_1} \left( U_{Y,j} - \bar{U}_Y \right)^2 \ \text{and} \ \widehat{\sigma}^2 := \frac{1}{n_1} \widehat{\sigma}_X^2 + \frac{1}{m_1} \widehat{\sigma}_Y^2. \quad (4)$$

Now we have completed the description of our studentized cross U-statistic $\overline{\text{xMMD}}^2 = \widehat{\text{xMMD}}^2 / \widehat{\sigma}$, and the resulting test $\Psi$ in (3). The asymptotic validity of the xMMD test is guaranteed by Theorem 15 that establishes the asymptotic normality of $\overline{\text{xMMD}}^2$ under the null.

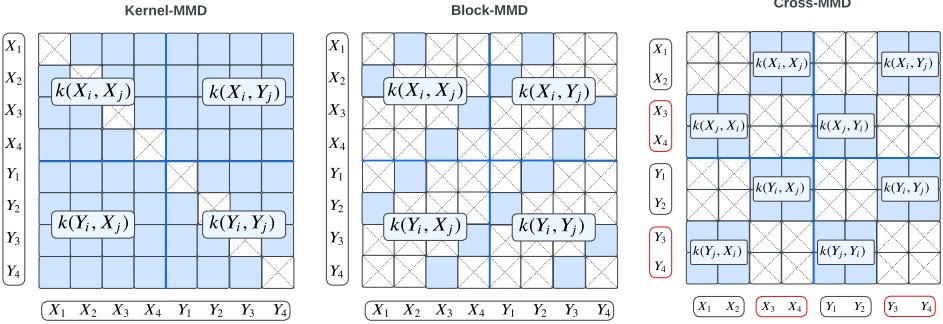

Figure 2: The figures visually illustrate the main differences in computing the usual quadratic-time kernel-MMD statistic (left), block-MMD (center) statistic, and our new cross-MMD statistic. In particular, the quadratic-time kernel-MMD statistic considers all pairwise kernel evaluations, with the exception of the diagonal terms. For block-MMD, we obtain the statistic by partitioning the data into several disjoint blocks; and then taking the average of the kernel-MMD statistic calculated over these disjoint blocks. Finally, our cross-MMD statistic first splits the data into two disjoint parts (red and black), and then uses the pairwise kernel evaluations with data from different splits. Interestingly, the observation pairs included by our cross-MMD statistic are exactly complementary to those included by the block-MMD statistic.

**Remark 2** (Computational Complexity). The overall cost of computing the statistic $\widehat{\text{x}\text{MMD}}^2$ is $\mathcal{O}\left((n+m)^2\right)$, and in particular, both $\widehat{\text{xMMD}}^2$ and $\widehat{\sigma}$ have quadratic complexity. To see this, note that $\widehat{\text{xMMD}}^2$ can be expanded into $\langle \widehat{\mu}_1, \widehat{\mu}_2 \rangle_k + \langle \widehat{\nu}_1, \widehat{\nu}_2 \rangle_k - \langle \widehat{\mu}_1, \widehat{\nu}_2 \rangle_k - \langle \widehat{\nu}_1, \widehat{\mu}_2 \rangle_k$. Each of these terms can be computed in $\mathcal{O}\left((n+m)^2\right)$. Similarly, each term in the summations defining $\widehat{\sigma}_X^2$ and $\widehat{\sigma}_Y^2$ also require $\mathcal{O}\left((n+m)^2\right)$ computation, implying that the $\widehat{\sigma}$ also has $\mathcal{O}((n+m)^2)$ complexity.

**Remark 3.** To simplify notation in what follows, we denote $m$ as $m_n$, where $m_n$ is some unknown nondecreasing sequence such that $\lim_{n\to\infty} m_n = \infty$. This still permits $m, n$ to be separate quantities growing to infinity at potentially different rates, but it allows us to index the sequence of problems with the single index $n$ (rather than $m, n$). We will use $k_n$, $d_n$, $\mathcal{X}_n$, $P_n$ and $Q_n$ to indicate that quantities could (but do not have to) change as $n$ increases, and drop the subscript when they are fixed. Furthermore, unless explicitly stated, we will focus on the balanced splitting scheme, i.e., $n_1 = \lfloor n/2 \rfloor$ and $m_1 = \lfloor m/2 \rfloor$ in what follows, because we currently see no apriori reason to split asymmetrically.

### 2.1 Gaussian limiting distribution under the null hypothesis

As shown in Figure 1, the empirical distribution of $\widehat{\text{x}\text{MMD}}^2$ resembles a standard normal distribution for various choices of $m$, $n$ and dimension $d$ under the null. In this section, we formally prove this statement. Recalling the mean embedding $\mu$ from (1), define

$$\bar{k}(x, y) := \langle k(x, \cdot) - \mu, k(y, \cdot) - \mu \rangle_k. \tag{5}$$

**Theorem 4.** *Suppose that $k$ and $P$ do not change with $n$. If $0 < \mathbb{E}_P[\bar{k}(X, X')^4] < \infty$ for $X, X' \overset{i.i.d.}{\sim} P$, then $\widehat{\text{x}\text{MMD}}^2 \overset{d}{\longrightarrow} N(0, 1)$.*

We next present a more general result that implies Theorem 4.

**Theorem 5.** *Suppose $P$ is fixed, but the kernel $k_n$ changes with $n$. If*

$$\lim_{n\to\infty} \frac{\mathbb{E}_P[\bar{k}_n(X_1, X_2)^4]}{\mathbb{E}_P[\bar{k}_n(X_1, X_2)^2]^2}\left(\frac{1}{n} + \frac{1}{m_n}\right) = 0, \quad and \quad \lim_{n\to\infty} \frac{\lambda_{1,n}^2}{\sum_{l=1}^{\infty} \lambda_{l,n}^2} \text{ exists}, \tag{6}$$

*where $(\lambda_{l,n})_{l=1}^{\infty}$ denote the eigenvalues of $\bar{k}$ introduced in (16), then we have $\widehat{\text{x}\text{MMD}}^2 \overset{d}{\longrightarrow} N(0, 1)$.*

It is easy to check that condition (6) is trivially satisfied if the kernels $\{k_n : n \geq 1\}$ are uniformly bounded by some constant; prominent examples are the Gaussian or Laplace kernel with a sample size

dependent bandwidth. Thus, the above condition really exists to handle unbounded kernels and heavy-tailed distributions. To motivate this requirement, we recall Bentkus and Götze (1996) (see Fact 11 in Appendix A) who proved a studentized CLT for i.i.d. random variables in a triangular array setup: $W_{1,n}, W_{2,n}, \ldots, W_{n,n} \overset{\text{i.i.d.}}{\sim} P_n$. Define $V_n = \sqrt{n} \sum_{i=1}^{n} W_{i,n} / \sqrt{\sum_i (W_{i,n} - \bar{W}_n)^2}$ where $\bar{W}_n = (\sum_i W_{i,n})/n$. They showed that a sufficient condition for the asymptotic normality of $V_n$ is that $\lim_{n\to\infty} \mathbb{E}_{P_n}[W_{1,n}^3] / \sqrt{\mathbb{E}_{P_n}[W_{1,n}^2]^3 n} = 0$. (This last condition is trivially true if $P_n$ does not change with $n$, meaning that the triangular array setup is irrelevant and $W_{1,n}$ can be replaced by $W_1$.)

Our requirement is slightly stronger: condition (6) with $\bar{k}_n(X_1, X_2)$ replaced by $W_{1,n}$ implies the previous condition of Bentkus and Götze (1996) (details in Remark 12 in Appendix A). We need this stronger condition, because the terms in the definition of $\widehat{\text{xMMD}}^2$ are not i.i.d. (indeed, not even independent), and thus we cannot directly apply the result of Bentkus and Götze (1996). Instead, we take a different route by first conditioning on the second half of data $(\mathbb{X}_2, \mathbb{Y}_2)$, then showing the conditional asymptotic normality of the standardized $\widehat{\text{xMMD}}^2$ (i.e., divided by conditional standard deviation instead of empirical), and finally showing that the ratio of conditional and empirical standard deviations converge in probability to 1 (see Appendix B).

Finally, we note that the result of Theorem 5 can be further generalized in several ways: **(i)** instead of a fixed $P$ and changing $k_n$, we can consider a sequence of pairs $\{(P_n, k_n) : n \geq 1\}$ changing with $n$, **(ii)** we can let $P_n \in \mathcal{P}_n^{(0)}$, for a class of distributions changing with $n$, and obtain the Gaussian limit uniformly over all elements of $\mathcal{P}_n^{(0)}$, and finally, **(iii)** the moment requirements on $\bar{k}_n$ stated in condition in (6) can also be slightly weakened. We state and prove this significantly more general version of Theorem 5 in Appendix B.

**Remark 6.** In the statement of the two theorems of this section, the splits $(\mathbb{X}_1, \mathbb{Y}_1)$ and $(\mathbb{X}_2, \mathbb{Y}_2)$ are assumed to be drawn i.i.d. from the same distribution $P$. However, a closer look at the proof of Theorem 5 indicates that the conclusions of the above two theorems hold even when the two splits are independent and drawn i.i.d. from possibly different distributions; that is $(\mathbb{X}_1, \mathbb{Y}_1)$ and $(\mathbb{X}_2, \mathbb{Y}_2)$ are independent of each other and drawn i.i.d. from distributions $P_1$ and $P_2$ respectively, with $P_1 \neq P_2$. In particular, under this more general condition, the asymptotic normality of $\overline{\text{xMMD}}^2$ still holds, and the resulting test $\Psi$ still controls the type-1 error at the desired level. This may be useful for two-sample testing in settings where the entire set of data is not i.i.d., but two different parts of the data were collected in two different situations. The usual MMD can also handle such scenarios by using a subset of permutations that do not exchange the data across the two situations.

## 2.2 Consistency against fixed and local alternatives

Here, we show that the xMMD test $\Psi$ introduced in (3) is consistent against a fixed alternative and also has minimax rate-optimal power against smooth local alternatives separated in $L^2$ norm.

We first show that analogous to Theorem 4, xMMD is consistent against fixed alternatives.

**Theorem 7.** *Suppose $P, Q, k$ do not change with $n$, and $P \neq Q$. If $k$ is a characteristic kernel satisfying $0 < \mathbb{E}_P[\bar{k}(X_1, X_2)^4] < \infty$, and $0 < \mathbb{E}_Q[\bar{k}(Y_1, Y_2)^4] < \infty$, then the xMMD test is consistent, meaning it has asymptotic power $1$.*

The moment conditions required above are mild, and are satisfied trivially, for instance, by bounded kernels such as the Gaussian kernel. The "characteristic" condition is also needed for the consistency of the usual MMD test (Gretton et al., 2012a), and is also satisfied by the Gaussian kernel.

Recalling Remark 1, we next consider the more challenging setting where $d_n, k_n$ can change with $n$, and $(P_n, Q_n)$ can vary within a class $\mathcal{P}_n^{(1)} \subset \mathcal{P}(\mathcal{X}_n) \times \mathcal{P}(\mathcal{X}_n)$ that can also change with $n$. We present a sufficient condition under which the xMMD test $\Psi$ is consistent *uniformly* over $\mathcal{P}_n^{(1)}$. Define $\gamma_n := \text{MMD}(P_n, Q_n)$, which is assumed nonzero for each $n$ but could approach zero in the limit.

**Theorem 8.** *Let $\{\delta_n : n \geq 2\}$ denote any positive sequence converging to zero. If*

$$\lim_{n\to\infty} \sup_{(P_n, Q_n) \in \mathcal{P}_n^{(1)}} \frac{\mathbb{E}_{P_n, Q_n}[\widehat{\sigma}^2]}{\delta_n \gamma_n^4} + \frac{\mathbb{V}_{P_n, Q_n}(\widehat{\text{xMMD}}^2)}{\gamma_n^4} = 0, \text{ where } \mathbb{V} \text{ denotes variance,} \quad (7)$$

*then* $\lim_{n\to\infty} \sup_{(P_n,Q_n)\in\mathcal{P}_n^{(1)}} \mathbb{E}_{P_n,Q_n}[1 - \Psi(\mathbb{X},\mathbb{Y})] = 0$, *meaning the* xMMD *test is consistent.*

Note that while *any* sequence $\{\delta_n\}$ converging to zero suffices for the general statement above, the condition (7) is easiest to satisfy for slowly decaying $\delta_n$, such as $\delta_n = 1/\log\log n$ for instance.

The sufficient conditions for consistency of $\Psi$ stated in terms of $\widehat{\sigma}$ and $\widehat{\text{xMMD}}^2$ in (7) can also be translated into equivalent conditions on the kernel function $k_n$, similar to (6), and we present the details in Appendix C. Importantly, if $P_n, Q_n, d_n$ are fixed and $k_n$ is bounded, then both $\mathbb{E}[\widehat{\sigma}^2]$ and $\mathbb{V}(\widehat{\text{xMMD}}^2)$ are $O(1/n)$, and $\gamma_n$ is a constant, so the condition is trivially satisfied, and in fact the above condition is even weaker than the fourth-moment condition of the previous theorem.

## 2.3 Minimax rate optimality against smooth local alternatives

We now apply the general result of Theorem 8 to the case where the distributions $P_n$ and $Q_n$ admit Lebesgue densities $p_n$ and $q_n$ that lie in the order $\beta$ Sobolev ball for some $\beta > 0$, defined as $\mathcal{W}^{\beta,2}(M) := \{f : \mathcal{X} \to \mathbb{R} \mid f \text{ is a.s. continuous, and } \int (1+\omega^2)^{\beta/2}\|\mathcal{F}(f)(\omega)\|^2 dw < M < \infty\}$. Formally, we define the null and alternative class of distributions as follows:

$$\mathcal{P}_n^{(0)} = \{P \text{ with density } p : p \in \mathcal{W}^{\beta,2}(M)\}, \quad \text{and}$$
$$\mathcal{P}_n^{(1)} = \{(P,Q) \text{ with densities } p,q \in \mathcal{W}^{\beta,2}(M) : \|p-q\|_{L^2} \geq \Delta_n\},$$

for some sequence $\Delta_n$ decaying to zero. In particular, we assume that under $H_0$, $P_n = Q_n$ and $P_n \in \mathcal{P}_n^{(0)}$, while under $H_1$, we assume that $(P_n, Q_n) \in \mathcal{P}_n^{(1)}$.

Our next result shows that for suitably chosen scale parameter, the xMMD test $\Psi$ with the Gaussian kernel is minimax rate-optimal for the above class of local alternatives. For simplicity, we state this result with $n = m$, noting that the result easily extends to the case when there exist constants $0 < c \leq C$, such that $c \leq n/m \leq C$.

**Theorem 9.** *Consider the case when $n = m$, and let $\{\Delta_n : n \geq 1\}$ be a sequence such that $\lim_{n\to\infty} \Delta_n n^{2\beta/(d+4\beta)} = \infty$. On applying the* xMMD *test $\Psi$ with the Gaussian kernel $k_{s_n}(x,y) = \exp(-s_n\|x-y\|_2^2)$, if we choose the scale as $s_n \asymp n^{4/(d+4\beta)}$, then we have*

$$\lim_{n\to\infty} \sup_{P_n \in \mathcal{P}_n^{(0)}} \mathbb{E}_{P_n}[\Psi(\mathbb{X},\mathbb{Y})] \leq \alpha \quad \text{and} \quad \lim_{n\to\infty} \inf_{(P_n,Q_n)\in\mathcal{P}_n^{(1)}} \mathbb{E}_{P_n,Q_n}[\Psi(\mathbb{X},\mathbb{Y})] = 1. \quad (8)$$

The proof of this statement is in Appendix C, and it follows by verifying that the conditions required by Theorem 8 are satisfied for the above choices of $\Delta_n$ and $s_n$.

**Remark 10.** Li and Yuan (2019, Theorem 5 (ii)) showed a converse of the above statement: if $\lim_{n\to\infty} \Delta_n n^{2\beta/(d+4\beta)} < \infty$, then there exists an $\alpha \in (0,1)$ such that any asymptotically level $\alpha$ test $\widetilde{\Psi}$ must have $\lim_{n\to\infty} \inf_{(P,Q)\in\mathcal{P}(\Delta_n)} \mathbb{E}_{P,Q}[\widetilde{\Psi}(\mathbb{X},\mathbb{Y})] < 1$. Hence, the sequence of $\{\Delta_n : n \geq 1\}$ used in Theorem 9 represents the smallest $L_2$-deviations that can be detected by any test, and (8) shows that our xMMD test $\Psi$ can detect such changes, establishing its minimax rate-optimality.

## 3 Experiments

We now present experimental validation of the theoretical claims of the previous section. In particular, our experiments demonstrate that **(i)** the limiting null distribution of $\widehat{\overline{\text{xMMD}}}^2$ is $N(0,1)$ under a wide range of choices of dimension $d$, sample sizes $n, m$ and the kernel $k$, and **(ii)** the power of our xMMD test is competitive with the kernel-MMD permutation test. We now describe the experiments in more detail. Additional experimental results are reported in Appendix E.

**Limiting null distribution of $\widehat{\overline{\text{xMMD}}}^2$.** We showed in Theorem 15 that the statistic $\widehat{\overline{\text{xMMD}}}^2$ has a limiting normal distribution under some mild assumptions. We empirically test this result when $\mathbb{X}$ and $\mathbb{Y}$ are drawn from $N(\mathbf{0}, I_d)$ with $\mathbf{0}$ denoting the all-zeros vector in $\mathbb{R}^d$, and in particular, study the effects of (i) dimension: $d = 10$ versus $d = 500$, (ii) skewness of the samples: $n/m = 1$ versus $n/m = 0.1$, and (iii) choice of kernel: Gaussian versus Quadratic, both with data-dependent scale parameters using *median* heuristic.

As shown in the first row of Figure 3, the distribution of $\widehat{\bar{\text{x}}\text{MMD}}^2$ is robust to all these effects, and is close to $N(0, 1)$ in all cases. In contrast, the distribution of the kernel-MMD statistic scaled by its empirical standard deviation (obtained using 200 bootstrap samples) in the bottom row of Figure 3 shows strong changes with these parameters. We present additional figures and details of the implementation in Appendix E.

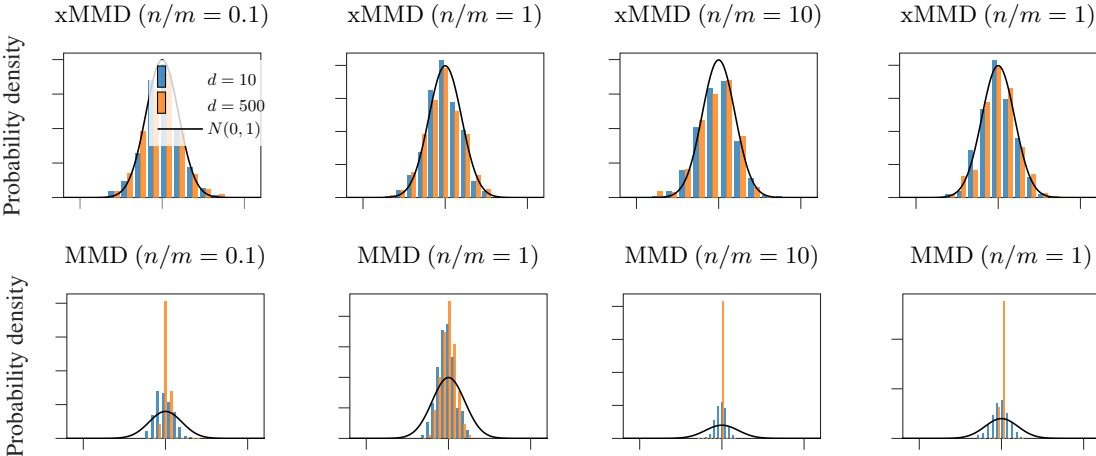

Figure 3: The first two columns show the null distribution of the $\widehat{\bar{\text{x}}\text{MMD}}^2$ statistic (top row) and the $\widehat{\text{MMD}}^2$ statistic scaled by its empirical standard deviation (bottom row) using the Gaussian kernel with scale-parameter chosen using the median heuristic. The last two columns show the null distribution for the two statistics using the Quadratic kernel with scale parameter chosen using the median heuristic. The figures demonstrate that the null distribution of $\widehat{\text{MMD}}^2$ changes significantly with dimension ($d$), the ratio $n/m$ and the choice of the kernel, unlike our proposed statistic.

**Evaluation of the power of $\Psi$.** For $d \geq 1$ and $j \leq d$, let $a_{\epsilon,j}$ denote the element of $\mathbb{R}^d$ with first $j$ coordinates equal to $\epsilon$, and others equal to 0. We consider the two-sample testing problem with $P = N(\mathbf{0}, I_d)$ $Q = N(a_{\epsilon,j}, I_d)$ for different choices of $\epsilon$ and $d$ and $j$. We compare the performance of our proposed test $\Psi$ with the kernel-MMD permutation test, implemented with $B = 200$ permutations, and plot the power-curves (using 200 trials) in Figure 4. We also propose a heuristic for predicting the power of the permutation test (denoted by $\rho_{\text{perm}}$) using the power of $\Psi$ (denoted by $\rho_\Psi$) as follows (with $\Phi$ denoting the standard normal cdf, and $z_\alpha$ its $\alpha$-quantile):

$$\widehat{\rho}_{\text{perm}} = \Phi\left(z_\alpha + \sqrt{2}\left(\Phi^{-1}(\rho_\Psi) - z_\alpha\right)\right). \tag{9}$$

This heuristic is motivated by the power expressions derived by Kim and Ramdas (2020) for the problems of one-sample Gaussian mean and covariance testing (we discuss this further in Appendix E). The term $\sqrt{2}$ in the above expression quantifies the price to pay for sample-splitting. As shown in Figure 4, this heuristic gives us an accurate estimate of the power of the kernel-MMD permutation test, without incurring the computational burden.

We now use ROC curves to compare the tradeoff between type-I and type-II errors for the usual MMD, linear and block-MMDs with our $\widehat{\bar{\text{x}}\text{MMD}}^2$. We use the same distributions $P = N(\mathbf{0}, I_d)$ and $Q = N(a_{\epsilon,j}, I_d)$ as before, and plot results for $(d, j, \epsilon) \in \{(10, 5, 0.2), (100, 20, 0.15), (500, 100, 0.1)\}$ in Figure 5. Due to sample splitting, the tradeoff achieved by our proposed statistic is slightly worse than that of $\widehat{\text{MMD}}^2$, but significantly better than other computationally efficient variants of kernel-MMD statistics. More details about the implementation are presented in Appendix E.

## 4 Conclusion and future work

We proposed a variant of the kernel-MMD statistic, called cross-MMD, based on the ideas of sample-splitting and studentization, and showed that it has a standard normal limiting null distribution. Using

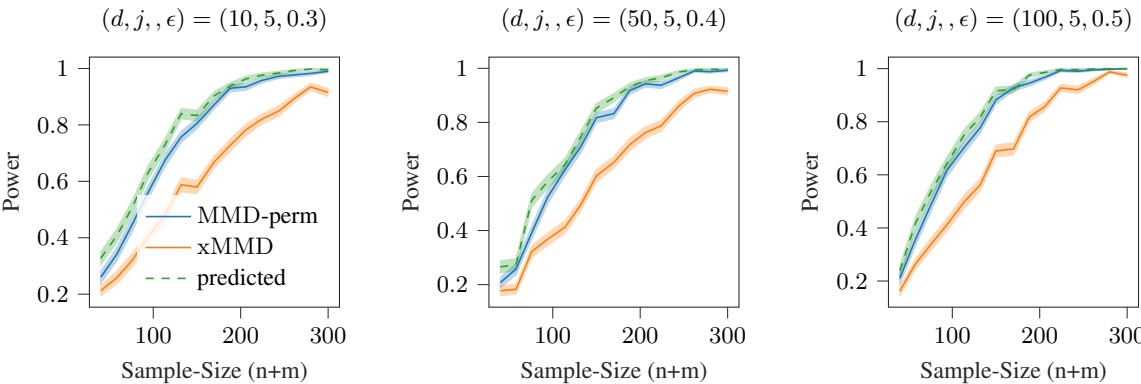

Figure 4: Curves showing the variation in power versus sample-size for the xMMD test and the kernel-MMD permutation test. $\mathbb{X}$ are drawn from $N(\mathbf{0}, I_d)$ i.i.d. and $\mathbb{Y}$ is drawn from $N(a_{\epsilon,j}, I_d)$ where $a_{\epsilon,j}$ is obtained by perturbing the first $j \leq d$ coordinates of $\mathbf{0}$ by $\epsilon$, the kernel used is the Gaussian kernel with scale parameter chosen via the median heuristic. The dashed curve shows the predicted power of the kernel-MMD permutation test using the heuristic defined in (9).

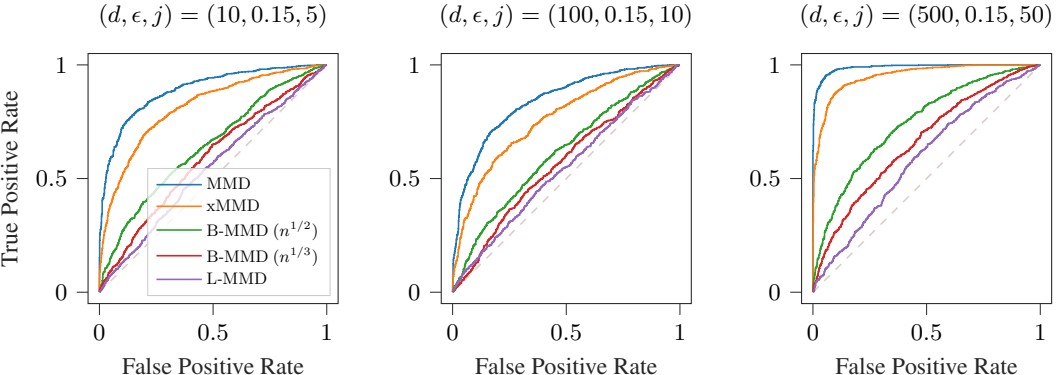

Figure 5: ROC curves highlighting the trade-off between type-I and type-II errors achieved by the MMD, cross-MMD, batch-MMD with batch sizes $n^{1/2}$ and $n^{1/3}$, and linear-MMD statistics. In all the figures, we use $n = m = 200$.

this key result, we introduced a permutation-free (and hence computationally efficient) MMD test for the two-sample problem. Experiments indicate that the power achieved by our test is within a $\sqrt{2}$-factor of the power of the kernel-MMD permutation test (that requires recomputing the statistic hundreds of times). In other words, our results achieve the following favorable tradeoff: *we get a significant reduction in computation at the price of a small reduction in power*.

Sejdinovic et al. (2013) establish in some generality that distance-based two-sample tests (like the energy distance (Székely and Rizzo, 2013)) can be viewed as kernel-MMD tests with a particular choice of kernel $k$. Hence our results are broadly applicable to distance-based statistics as well. Since two-sample testing and independence testing can be reduced to each other, it is an interesting direction for future work to see if the ideas developed in our paper can be used for designing permutation-free versions of kernel-based independence tests like HSIC (Gretton et al., 2007) or distance covariance (Székely and Rizzo, 2009; Lyons, 2013). Our techniques seem to rely on the specific structure of two-sample U-statistics of degree 2. Extending these to more general U-statistics of higher degrees is another important question for future work. A final question is to figure out whether it is possible to achieve minimax optimal power using a sub-quadratic time test statistic. One potential approach would be to work with a kernel approximated by random Fourier features (Rahimi and Recht, 2007; Zhao and Meng, 2015). Depending on the number of random features, our test statistic can be computed in sub-quadratic time and it would be interesting to see whether the resulting test can still be minimax optimal in power. We leave this important question for future work.

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
