# OpenReview forum: "A permutation-free kernel two-sample test"
_NeurIPS.cc/2022/Conference — NeurIPS 2022 Accept_

### Official Review · Reviewer_qgsP · 2022-07-04

**Rating:** 7
**Confidence:** 4
**Soundness:** 4 excellent
**Presentation:** 4 excellent
**Contribution:** 3 good

**Summary:**

The paper proposes a variant of the quadratic time MMD test, whose null distribution is a standard normal, and whose statistic is computed based on splitting the available samples (e.g. in half).
This, while reducing the test-power, allows for easy test threshold estimation, and in particular avoid the permutation-based approach of the original quadratic time MMD test.
The authors formally derive this null distribution, and proove a number of properties of the resulting test, including consistency against fixed local alternatives, and finally a minimax rate-optimal power against local alternatives (Gaussian kernel only).
A number of experiments show that the theory translates into practice.


EDIT after rebuttal. The authors have provided an excellent and careful reply to all of my concerns. I have increased my score and think this is a great paper which should be accepted.

**Questions:**

See my points on the computational claims above.
I strongly suggest to add the linear time test baseline mentioned above.

**Limitations:**



**Strengths And Weaknesses:**

A great paper that tackles a worthwhile problem in a creative manner, and is thoroughly executed, in particular theory wise.
I am not sure about the practical impact of the paper though, because hundredfold computational improvements stated by the authors are somewhat academic, and do not hold true against careful implementations. In practice, I suspect the different to be much smaller, which then raises the question whether the loss in test power can really be justified.
The paper in my opinion also misses an important baseline test.


Originality:
The idea of building an MMD test with a normal null distribution has been around since the original MMD test has been first described.
Back then, however, the test statistic was linear time which meant the test wasn't as powerful.
It is great that the authors found a way to achieve a standard normal null using a quadratic time test, and I consider this very original work.

Quality:
The paper is executed on a very high technical level, all claims have substantial proofs.
I in particular appreciate the generalization of Theorem 6 to the case where the kernel is allowed to change. Great work.
The minimax optimality is great too, though only established for Gaussian kernels (this should be stated in the abstract, where it appears that this is a general results)

Experimentally, I think one baseline that is clearly missing is the linear time test with learned features, described in https://arxiv.org/pdf/1605.06796.pdf. The paper reports experiments where a linear time test performs *better* than the standard quadratic time MMD. So this is highly relevant for this paper as well. In fact, I think the paper should not be published without these baselines.

On a side note, the authors do not experimentally demonstrate that their test has a well calibrated type 1 error for a given threshold, which I think would be a nice addition.

Another side note, the authors could run a standard goodness of fit test on the results in Fig 2, to test for normality.

Clarity:
Clearly written throughout.

Significance:
From a theoretical and methodological perspective, this work is significant, as it improves the ever growing set of tools around MMD tests.

However, I am not so sure about practical impact, and this is my main critique of the paper.
1. The authors argue that the permutation computations requires for the original test are prohibitive. In practice however, these are trivially parallelizable and not time critical. The idea behind the quadratic time MMD has always been "the best possible test for a given set of samples". Making that faster is useful, but not at the cost of sacrificing power, as in practice maximum power is usually what is wanted.
2. A more severe point is that the authors argue that the permutation test is in fact hundreds of times slower than their proposed test. This is only true asymptotically. In practice, permutation tests can be implemented in an extremely efficient manner by precomputing the kernel matrix, and then using memory sequential read operations to compute the permutations. As also mentioned above, this can be easily parallelized. E.g. https://arxiv.org/abs/1611.04488 reported that a naive python implementation (using precomputed kernel matrices) can be made >10x faster using these simple strategies, and before parallelization, plus another 5-10x using multithreaded implementations. In summary, with a little implementation effort, the computational gain (in wallclock time, with efficient implementations and parallel hardware), will surely not be hundredfold.

---

> ### Author Response · Authors · 2022-08-02
> **Response to Reviewer qgsP (1)**
>
> We thank the reviewer for their feedback and respond to their questions below.
>
> > Experimentally, I think one baseline that is clearly missing is the linear time test with learned features, described by Jitkrittum et al. (2016). The paper reports experiments where a linear time test performs better than the standard quadratic time MMD. So this is highly relevant for this paper as well. In fact, I think the paper should not be published without these baselines.
>
>
>  We thank the reviewer for bringing this paper to our attention. As suggested by the reviewer, we have included comparisons with the two tests (ME and SCF tests) introduced in this paper in Appendix~F.1. Surprisingly, when $n$ is small ($\leq 250$), our cross-MMD test with bandwidth selected via the median heuristic outperforms the ME and SCF tests; both in terms of power and control of type-I error.
>
> While these tests are certainly an important alternative to permutation based tests, we would like to emphasize the following points that distinguish our approach.
>
> 1. The ME and SCF tests require the kernel to be uniformly bounded, whereas our test requires only mild moment conditions that are even satisfied by unbounded kernels if the underlying distributions are not too heavy-tailed. Futhermore, the ME and SCF tests have several tuning parameters: number of features $J$, $\{v_1, \ldots, v_J\}$, bandwidth, step-size for gradient ascent etc.. In practice, $J$ is usually set to $5$, and the other parameters are selected by solving a $Jd + 1$ dimensional optimization problem via gradient ascent. While each step of gradient ascent had linear in $n$ complexity, the number of steps needed may be large for higher dimensions. We observed this in our experiments, where from small $n$ and large $d$, the optimization overhead was noticeable, and made the ME and SCF tests take longer than cross-MMD.
>
> 2. More importantly, the ME and SCF tests are only valid in the 'low-dimensional setting': fixed $d$ and $J$, with $n \to \infty$. In the high dimensional setting, when $(d, n) \to \infty$,  the limiting null distribution may no longer be $\chi^2(J)$. We observed this in some simulations (e.g., Fig.10 in Appendix  F.1) when we fixed $d=100$, and varied $n$ from $50$ to $250$. This results in the following practical issue: given a problem with $n=500$ and $d=200$, how should one calibrate the threshold for those tests?
>
>     Our proposed test does not suffer from this, because in both high and low dimensional settings, our statistic has the same limiting distribution. This is a significant practical advantage of our cross-MMD test over ME and SCF tests.
>
>
> 3. In the regime where the number of features, $J$, is allowed to increase with $n$, we expect that the resulting ME and SCF tests may have low power (for small regularization parameter $\gamma_n$). This is because, the test statistic $\hat{\lambda}_n$ used by ME and SCF tests is similar to Hotelling's T^2 statistic, for which Bai and Sarandasa (1996) characterized the asymptotic power in this regime. In particular,  their Theorem 2.1 implies that the power of the $T^2$ test grows slowly with $n$, especially when $J/n \approx 1$.
>
> > On a side note, the authors do not experimentally demonstrate that their test has a well calibrated type 1 error for a given threshold, which I think would be a nice addition.
>
> * We have included two figures showing the type-I error in low and high dimensional settings in Appendix F.2
>
> > Another side note, the authors could run a standard goodness of fit test on the results in Fig 2, to test for normality.
>
> * Thanks  for this suggestion. We have included it in Appendix F.2.

---

> > ### Author Response · Authors · 2022-08-02
> > **Response to Reviewer qgsP (2)**
> >
> >
> > > The authors argue that the permutation computations requires for the original test are prohibitive. In practice however, these are trivially parallelizable and not time critical. The idea behind the quadratic time MMD has always been "the best possible test for a given set of samples". Making that faster is useful, but not at the cost of sacrificing power, as in practice maximum power is usually what is wanted.
> >
> > While we understand the reviewer's opinion, we note that there do exist several works proposing computationally cheaper versions of the MMD  (including the baselines suggested by the reviewer), as well as papers studying the computational-statistical tradeoffs of kernel based tests. These tests go back almost a decade: thus, it seems that the literature has been interested in finding computationally cheaper alternatives to the MMD, and we believe ours is the first which achieves the minimax rate-optimal power. While we agree that this would not displace the full MMD from being the gold standard, it does possibly have a place as an important contribution in the surrounding literature, providing a complementary alternative. Beyond the methodology, we feel like the idea behind the test and the surrounding theory, is quite interesting and may be applicable to other settings.
> >
> >
> > > A more severe point is that the authors argue that the permutation test is in fact hundreds of times slower than their proposed test. This is only true asymptotically. In practice, permutation tests can be implemented in an extremely efficient manner by precomputing the kernel matrix, and then using memory sequential read operations to compute the permutations. As also mentioned above, this can be easily parallelized. E.g. Sutherland et al.(2017) reported that a naive python implementation (using precomputed kernel matrices) can be made >10x faster using these simple strategies, and before parallelization, plus another 5-10x using multithreaded implementations. In summary, with a little implementation effort, the computational gain (in wallclock time, with efficient implementations and parallel hardware), will surely not be hundredfold.
> >
> > We thank the reviewer for pointing out this efficient implementation of the kernel-MMD statistic to us. We have replaced the word 'hundredfold' with 'significant' in our abstract (since the gain could still be tenfold, depending on the number of permutations and implementation details).
> >
> > Additionally, we would like to note the following:
> > * Our solution is simple and appealing, and may be most useful in complementary tasks (as compared to the implementation of Sutherland et al. 2017); such as when we don't have access to parallel computing infrastructure, or if the data or number of permutations is very large.
> > * More importantly, our ideas also directly extend to other U-statistics as we discuss in Appendix D. Hence, they can be used in cases where we may not yet have  an efficient implementation similar to that proposed for the kernel MMD statistic by Sutherland et al. (2017).
> >
> > **References**
> > 1. Jitkrittum et al. "Interpretable distribution features with maximum testing power." Neurips, 2016.
> > 2. Sutherland et al. "Generative models and model criticism via optimized maximum mean discrepancy." ICLR 2017.
> > 3. Bai and Saranadasa. "Effect of high dimension: by an example of a two sample problem." Statistica Sinica, 1996.

---

### Official Review · Reviewer_ocYj · 2022-07-10

**Rating:** 7
**Confidence:** 3
**Soundness:** 4 excellent
**Presentation:** 4 excellent
**Contribution:** 4 excellent

**Summary:**

The paper proposes and investigates an alternative estimator for maximum mean  discrepancy (MMD), a popular kernel-based two sample test. The benefit of the estimator is that it has a standard normal distribution under the null hypothesis of equal distributions. Thus, there is no need to perform costly permutation tests in order to get a p-value from the surrogate null distribution. The theoretical results go beyond this to include cases where the kernel is sample size dependent, as well as showing a mini-max optimality against local alternative hypotheses where the distributions are close in terms of a probability metric.

**Questions:**

Could cross-MMD be used to choose from amongst different kernel sizes and then use MMD for the test?


Suggestion: Equation (9) it could be made clear to the reader what $z_\alpha$ is.

**Limitations:**

Requiring larger sample sizes for computational ease may not be warranted in situations where data collection is costly.

**Strengths And Weaknesses:**

Strengths:
The paper is an original and important contribution that is well-written, having a comprehensive introduction and discussion of related work, goes into technical details with what appears to be rigorous theory. The paper is rather complete with key implications for future work on other distance or kernel based tests including tests of dependence.

Weaknesses:
The application and implication to some real scenarios is not completely discussed. In real cases where the data is collected (or will be collected) the sample sizes are fixed (or wouldn't want to be larger). In these cases, is the computational savings going to be meaningful?

---

> ### Author Response · Authors · 2022-08-02
> **Response to Reviewer ocYj**
>
> We thank the reviewer for their feedback, and answer their questions below.
>
> > The application and implication to some real scenarios is not completely discussed. In real cases where the data is collected (or will be collected) the sample sizes are fixed (or wouldn't want to be larger). In these cases, is the computational savings going to be meaningful?
>
> If the sample-size is small, then the computational gains over the permutation test would not be significant. However, in cases with limited computing power, with moderate to large sample sizes, the power-computational complexity tradeoff achieved by our cross-MMD test would be meaningful (as also demonstrated through simulations in our paper). We emphasize that we do not mean for our test to replace the MMD, it simply complements the MMD by providing a simple alternative that can be quickly implemented by a novice, and may be preferable in some (but definitely not all) settings.
>
>
> > Could cross-MMD be used to choose from amongst different kernel sizes and then use MMD for the test?
>
>  Thanks for this suggestion. Since the predicted power using cross-MMD seems to match the power of the MMD (at least, in the settings considered in our experiments), your idea could indeed be used for selecting the best of $M$ bandwidths. However, we note that this approach  would work well in practice if $M$ is an order smaller than $B$, the number of permutations to be used in the eventual test. (For example, $M=50$ and $B=500$ would be reasonable default choices.)
>
>
>
> > Suggestion: Equation (9) it could be made clear to the reader what  $z_{\alpha}$ is.
>
> Thanks for the suggestion. We now indicate in the updated version of the manuscript that it refers to the $\alpha$ quantile of a standard Gaussian.

---

> > ### Comment · Reviewer_ocYj · 2022-08-07
> > **Response to author changes**
> >
> > The authors have done a thorough job of responding to my comments and questions as well as the other reviewers. I have no other concerns at this time.

---

### Official Review · Reviewer_ja6F · 2022-07-11

**Rating:** 7
**Confidence:** 4
**Soundness:** 3 good
**Presentation:** 3 good
**Contribution:** 3 good

**Summary:**

This paper proposes a novel two-sample test based on the MMD test. This test has a closed-form of null distribution (a normal distribution), and the intuition for the resulting null distribution is that the proposed test can be thought of as performing a student's t-test.

**Questions:**

Please see above.

**Limitations:**

Please see above.

**Strengths And Weaknesses:**

Strength: It is creative to split the dataset into two pairs to studentize the original MMD test, therefore resulting to a traceable null distribution.
Weakness: Comparison with other permutation two-sample test which also has closed-form of the null distribution is missing. One example is the Friedman-Rafsky (FR) test[Friedman, 1979]. In fact, both the FR test and the proposed test have the same null distribution, I think the FR test would be a good baseline would be good to compare with. Besides, the price of deriving such an ''exact'' null distribution test is to decrease the power. Is that possible to have a sample complexity analysis regarding the Type II error of the proposed MMD variant compared with the original MMD test?
Friedman, Jerome H., and Lawrence C. Rafsky. "Multivariate generalizations of the Wald-Wolfowitz and Smirnov two-sample tests." The Annals of Statistics (1979): 697-717.

---

> ### Author Response · Authors · 2022-08-02
> **Response to Reviewer ja6F**
>
> We thank the reviewer for their feedback, and answer their questions below.
>
>
> > Comparison with other permutation two-sample test which also has closed-form of the null distribution is missing. One example is the Friedman-Rafsky (FR) test[Friedman, 1979]. In fact, both the FR test and the proposed test have the same null distribution, I think the FR test would be a good baseline would be good to compare with.
>
> * We thank the reviewer for this suggestion, and we have included comparisons of the FR test with cross-MMD in Appendix F.3.
> * We would also like to mention, that from a theoretical point of view, the FR (and similar tests based on geometric graphs)  suffer from the drawback of being powerless against any local alternatives at a $\mathcal{O}(n^{-1/2})$ distance in $L^2$ norm (ie $\|P-Q\|_2 \asymp n^{-1/2}$), as shown by Bhattacharya (2019). In contrast, our test has  power against sufficiently smooth local alternatives at $\mathcal{O}(n^{-1/2})$ as a consequence of Theorem 8. Indeed, that Theorem shows that our test achieves the minimax optimal rate (in the appropriate norm) for any $n,d,\beta$, unlike FR and related tests.
>
>
> > Besides, the price of deriving such an ''exact'' null distribution test is to decrease the power. Is that possible to have a sample complexity analysis regarding the Type II error of the proposed MMD variant compared with the original MMD test?
>
> Theorem 8 already contains a partial answer to your question; albeit written in a different form.
> To elaborate, in nonparametric testing, the power of a test is often analyzed by considering a sequence of alternatives that get increasingly closer to the null with sample-size $n$. Following this, Theorem 8 characterizes the minimum separation $\Delta_n$ that can be consistently detected by our test in terms of $n$, $d$ and smoothness $\beta$.
> Inverting the expression in Theorem 8, one can infer that: if given distributions $P$ and $Q$ are separated in the $L^2$ sense by some $\Delta>0$, then $\Omega(\Delta^{-(2 + d/(2\beta))})$ data-points suffice to achieve high detection power. This sample complexity is sufficient for both the usual MMD statistic and our cross-MMD statistic based test, and it is necessary in the sense that no other test can achieve a better sample complexity (making both tests minimax rate-optimal). Thus the only difference between the MMD and the cross-MMD is in the constant factor of about $\sqrt{2}$, and not in the dependence on $n$, $d$, smoothness $\beta$, or other problem dependent constants.
>
>
> **References**
> 1. Bhattacharya, Bhaswar B. "A general asymptotic framework for distribution‐free graph‐based two‐sample tests." Journal of the Royal Statistical Society: Series B (Statistical Methodology) 2019.

---

### Official Review · Reviewer_794f · 2022-07-12

**Rating:** 7
**Confidence:** 4
**Soundness:** 4 excellent
**Presentation:** 3 good
**Contribution:** 2 fair

**Summary:**

This proposes a variant of the kernel-MMD statistic, called cross-MMD, based on the ideas of sample splitting and studentization, and showed that it has a standard normal limiting null distribution. Using this key result, they introduced a permutation-free (and hence computationally efficient) MMD test with 100 times faster and a sacrifice in performance that is better than the factor sq(2) (expected due to splitting).

**Questions:**

Considering the current presentation of the results It is not clear if always sqrt(2) factor is always guaranteed. Please correct me if I misunderstood. For example In cases of Figure 3 for samplesizes of 50 to 100 is it true? Please try to discuss if there are cases where the test's performance is not sufficient (therefore the test is not recommended).

**Limitations:**

Authors are discussing sufficiently both limitations and perspectives. The weaknesses that I mentioned could also become part of this discussion.

**Strengths And Weaknesses:**

Strengths
-Well written paper. Nice and simple theory. No "wordy" sentences, No blurry experiments or non-understandable mathematical expressions. No overselling. Well done.

Weaknesses
- Focus on performance and time consumption but (MMD-related weakness) no information concerning interpretability in terms of Variable influence and how mmd can be used as an alternative to traditional approaches (multiple testing etc).
- The splitting scheme is always problematic in cases with low sample sizes (and the time gain is negligible) and studentization more risky.
- I see a contribution only in cases where extremely big datasets occur.

---

> ### Author Response · Authors · 2022-08-02
> **Response to Reviewer 794f**
>
> We thank the reviewer for their feedback, and answer their questions below.
>
>
> >  It is not clear if always $\sqrt(2)$ factor is guaranteed $\ldots$ In cases of Figure 3 for samplesizes of 50 to 100, is it true?
>
> The $\sqrt{2}$ factor is not guaranteed at small sample sizes; but in this regime, running the full permutation test itself is feasible (that is, not computationally prohibitive). It appears that our asymptotics do kick in at relatively small sample sizes, for example,  with $n \geq 150$ as shown in  Figure 3 of the manuscript. So the $\sqrt{2}$ factor is not a bad approximation for most problems.
>
> > Please try to discuss if there are cases where the test's performance is not sufficient (therefore the test is not recommended).
>
> Yes, there are  certain situations in which the cross-MMD test may not be suitable:
> * The most obvious situation is when computation is not a factor, either due to small sample-size or due to the available computing infrastructure. In such situtations the permutation test is certainly preferable.
> *  Our type-I error guarantees require certain mild conditions on the growth rate of the fourth moment of the kernel. These conditions are trivially satisfied for bounded kernels, such as the Gaussian kernel. But in situations where we wish to use unbounded kernels that may violate these requirements, permutation tests are preferable.

---

### Author Response · Authors · 2022-08-02
**Summary of Changes**

We thank the reviewers for their feedback. We have made the following changes to the manuscript:

1. We have added all the additional experimental results, and discussions related to the reviewers' questions in Appendix F of the supplementary.

2. We have highlighted the (minor) changes made to the main text in blue.

---

### Meta-Review · Area_Chair_g6po · 2022-08-26

**Recommendation:** Accept
**Confidence:** Certain

**Metareview:**

The paper proposes a variant of MMD test, which is asymptotically normal.  The proposed method is computed easily with data-splitting and studentization.  The paper provides a solid theoretical study about the consistency and minimax optimal rate for local alternatives.  The experimental results also show favorable results.
The proposed simple method for obtaining the asymptotic normality is a significant advance in the topic of MMD, which is a popular statistic for comparing two distributions but suffers from the complicated asymptotics in the previous variants.  The theoretical study is solid.  We believe the work is worth being presented in NeurIPS.

**Award:**

No

---

### Decision · Program_Chairs · 2022-09-14

Accept